# Development of a Combined Genetic Engineering Vaccine for Porcine Circovirus Type 2 and *Mycoplasma Hyopneumoniae* by a Baculovirus Expression System

**DOI:** 10.3390/ijms20184425

**Published:** 2019-09-09

**Authors:** Yu Tao, Gaojian Li, Wenqian Zheng, Jianhong Shu, Jian Chen, Fang Yang, Yuehong Wu, Yulong He

**Affiliations:** 1Department of Biopharmacy, College of Life Sciences and Medicine, Zhejiang Sci-Tech University, Hangzhou 310018, China (Y.T.) (G.L.) (W.Z.) (J.S.) (J.C.); 2Hangzhou Sino-science Gene Technology Co., Ltd., Hangzhou 310018, China

**Keywords:** baculovirus expression system, chimeric protein, combined vaccine, *Mycoplasma hyopneumoniae*, porcine circovirus type 2

## Abstract

*Mycoplasma hyopneumoniae* (Mhp) and porcine circovirus type 2 (PCV2) are the main pathogens for mycoplasmal pneumonia of swine (MPS) and post-weaning multisystemic wasting syndrome (PMWS), respectively. Infection by these pathogens often happens together and causes great economic losses. In this study, a kind of recombinant baculovirus that can display P97R1P46P42 chimeric protein of Mhp and the capsid (Cap) protein of PCV2 was developed, and the protein location was identified. Another recombinant baculovirus was constructed without tag proteins (EGFP, mCherry) and was used to evaluate the immune effect in experiments with BALB/c mice and domestic piglets. Antigen proteins P97R1P46P42 and Cap were expressed successfully; both were anchored on the plasma membrane of cells and the viral envelope. It should be emphasized that in piglet immunization, the recombinant baculovirus vaccine achieved similar immunological effects as the mixed commercial vaccine. Both the piglet and mouse experiments showed that the recombinant baculovirus was able to induce humoral and cellular responses effectively. The results of this study indicate that this recombinant baculovirus is a potential candidate for the further development of more effective combined genetic engineering vaccines against MPS and PMWS. This experiment also provides ideas for vaccine development for other concomitant diseases using the baculovirus expression system.

## 1. Introduction

*Mycoplasma hyopneumoniae* (Mhp) is the main pathogen of mycoplasmal pneumonia of swine (MPS), which is characterized by a wide distribution and high incidence but a low lethality. Mhp infection can destroy the cilium barrier of the respiratory tract and lead to secondary infections such as porcine circovirus type 2 (PCV2), porcine reproductive and respiratory syndrome (PRRS) virus, *Mycoplasma hyorhinis*, and *Mycoplasma flocculare* [1,2,3]. Among these pathogens, PCV2 is the causative agent of post-weaning multisystemic wasting syndrome (PMWS) and porcine circovirus-associated disease (PCVAD) [4]. The co-infection of Mhp and PCV2 can cause serious immunosuppression and increase mortality, which brings great economic losses to pig husbandry production worldwide [5,6].

Vaccination is still the main method to control diseases in production; live attenuated or inactivated vaccines are the main vaccines used for Mhp [7]. Some studies have shown that these infections cannot induce the immune system to produce significant levels of antibodies against antigens of Mhp and can only provide partial protection against MPS [5,6]. Genetically engineered vaccines against Mhp at the experimental stage have shown effective protection, but they are not yet on the market. Moreover, some signature antigens of Mhp have been identified and used for vaccine development, such as P97R1, P46, and P42, and genetically engineered vaccines based on these antigens have shown immune effects in animal experiments [8,9,10,11]. Capsid protein (Cap) is encoded by open reading frame 2 (ORF2) of PCV2 and has been identified as a major structural protein and antigen [12]. Cap has been expressed successfully with expression systems such as *Escherichia coli* (*E. coli*), yeast, and baculovirus, and some Cap-protein-based genetically engineered vaccines are now available [13,14,15]. Studies have found that combined Mhp and PCV2 vaccination is convenient and efficacious against co-infection, indicating that the development of combined vaccines for Mhp and PCV2 is feasible and advantageous [16,17].

Compared with the prokaryotic expression system, the eukaryotic expression system has more perfect modification of protein structure and is widely used in vaccine research and development. Studies have indicated that recombinant adenoviruses expressing antigens can induce effective immune protection and can be used for vaccine development [18,19]. In addition, the baculovirus protein expression system has been further developed in recent years and has become the focus of vaccine development [1,2,3]. With the assembly of recombinant baculovirus nucleocapsid, foreign proteins are synthesized and anchored to the cytoplasmic membrane, recombinant baculovirus envelope, and recombinant baculovirus core-capsid surface [20,21,22]. Plasmid pFastBac dual is an insect cell expression vector that can express two foreign genes simultaneously with the polyhedrin (pH) promoter and p10 promoter. Based on these characteristics, antigens forming different pathogens can be expressed by baculovirus simultaneously and act as genetically engineered vaccines with the antigens anchored to the baculovirus envelope.

In this study, chimeric protein P97R1P46P42 from Mhp and Cap protein from PCV2 were expressed and located successfully using the baculovirus expression system. The immunogenicity of this recombinant baculovirus was also evaluated by way of immunization performed in mice and piglets. The results demonstrated that the baculovirus expression system can be used as an effective strategy to develop combined vaccines against co-infection of Mhp and PCV2 and provide new ideas for the development of vaccines for other concomitant diseases.

## 2. Results

### 2.1. Vector Construction, Antigen Expression in Sf9 Cells, and Western Blot Identification

Two plasmids were constructed in this study: pFastBac dual-P97R1P46P42-mCherry-Cap-EGFP (Figure 1A) and pFastBac dual-P97R1P46P42-Cap (Figure 1B). Both the P97R1P46P42 and Cap genes were synthetic; plasmids were identified by sequencing, and bacmids were verified by PCR with M13 primers. The recombinant baculoviruses were named rvAc-P97R1P46P42-mCherry-Cap-EGFP and rvAc-P97R1P46P42-Cap. Sf9 cells infected with rvAc-P97R1P46P42-Cap were lysed and the virus was collected at the same time. For long-term storage and immunization, we have controlled the titer of the virus in 10^9^ PFU/mL by purification and concentration. Five primary antibodies (anti-P97R1, anti-P46, anti-P42, anti-Cap, and anti-6×His) were used to detect the expression of antigens in the cells and virus by Western blot (Figure 2).

### 2.2. Localization of Antigen Expression in Cells and Recombinant Baculovirus rvAc-P97R1P46P42-mCherry-Cap-EGFP

Sf9 cells infected with rvAc-P97R1P46P42-mCherry-Cap-EGFP were observed in different fields (white, green, blue) after being incubated at 27 °C for 72, 96, and 120 h (Figure 3). For further protein localization, indirect immunofluorescence assay was performed, and the results indicate that the two antigens (mCherry-tagged P97R1P46P42 and EGFP-tagged Cap protein) were directly located on the plasma membrane of infected Sf9 cells (Figure 4). To detect whether the antigens were displayed on the baculovirus envelope, the rvAc-P97R1P46P42-mCherry-Cap-EGFP and rvAc-dual recombinant baculoviruses were purified by sucrose gradient ultracentrifugation and visualized with immunogold electron microscopy (Figure 5). This result indicated that the recombinant baculovirus successfully displayed the two fluorescence-tagged antigens on the envelope.

### 2.3. Systemic Humoral and Cellular Immune Response of rvAc-P97R1P46P42-Cap Induction in Mice

To evaluate the humoral immune response of rvAc-P97R1P46P42-Cap induction in mice, the immunoglobulin G (IgG) level was determined by indirect ELISA, and the results are shown in Figure 6A–D. The level of anti-rP97R1 was statistically higher in sera from the rvAc-P97R1P46P42-Cap group than in those from the other groups at 35 and 42 days after immunization (DAI) (*p* < 0.001; Figure 6A). The sera from the rvAc-P97R1P46P42-Cap group had statistically higher (*p* < 0.001) levels of antibodies against rP46 and rP42 than did those from the other four groups at 14, 28, 35, and 42 DAI (Figure 6B,C). It is worth noting that the antibody level against Cap from the rvAc-P97R1P46P42-Cap group was significantly higher (*p* < 0.001) than that from the PBS (experimental group injected with phosphate-buffered saline, regarded as the control group), rvAc-dual, and Mhp CV groups at 28, 35, and 42 DAI, but lower than that from the PCV2 CV group (Figure 6D). Moreover, sera from the Mhp CV group did not react with all antigens (Figure 6A–D).

The cellular immune response induced by rvAc-P97R1P46P42-Cap was also evaluated, and the stimulation value of the rvAc-P97R1P46P42-Cap group was significantly higher (*p* < 0.001) than that of the other four groups at 35 and 42 DAI (Figure 6E). The concanavalin A control (positive control of the lymphocyte proliferation assay) worked efficiently, and its stimulation value was regarded as 100%.

To verify whether the antibodies were able to recognize native proteins, sera collected at 42 DAI were evaluated against Mhp 168 and PCV2 ZJ/C strain extract by ELISA. For the Mhp 168 strain, the antibody levels from the Mhp CV and rvAc-P97R1P46P42-Cap groups were significantly higher (*p* < 0.001 and *p* < 0.05, respectively) than those from the PBS, rvAc-dual, and PCV2 CV groups; for PCV2 ZJ/C, sera from the PCV2 CV and rvAc-P97R1P46P42-Cap groups had statistically higher (*p* < 0.01 and *p* < 0.001, respectively) antibody levels than those from the PBS, rvAc-dual, and Mhp CV groups (Figure 6F,G).

### 2.4. Immunogenicity in Piglets

To evaluate the immune efficacy of recombinant baculovirus induction in piglets, the humoral immune responses of commercial vaccines and recombinant baculovirus were determined by ELISA with the method described above (Figure 7A–D). Sera from CV and rvAc-P97R1P46P42-Cap groups at 28 DAI had statistically higher (*p* < 0.001) levels of antibodies against the four antigens than did those from the rvAc-dual group. The antibody level of the CV group was higher than that of rvAc-P97R1P46P42-Cap group in serum at 28 DAI.

The cellular immune responses induced by commercial vaccines and recombinant baculovirus were evaluated by analyzing the levels of interleukin 4 (IL-4) and interferon gamma (IFN-γ) in serum (Figure 7E,F). The IL-4 and IFN-γ levels of the CV and rvAc-P97R1P46P42-Cap groups were statistically higher (*p* < 0.001) than those of the rvAc-dual group. The IL-4 levels in sera from the rvAc-P97R1P46P42-Cap and CV groups ranged from 47.79 to 73.05 pg/mL and 69.12 to 77.21 pg/mL, respectively. The IFN-γ levels in sera from the rvAc-P97R1P46P42-Cap and CV groups ranged from 41.74 to 44.80 pg/mL and 56.63 to 60.45 pg/mL, respectively.

## 3. Discussion

Both MPS and PMWS are important diseases in the swine industry, and the co-infection of Mhp and PCV2 is one of the most important etiological contributors of the porcine respiratory disease complex [23]. At present, the main method of prevention is by vaccine, and the focus has gradually shifted to a combined vaccine against the two diseases as the same time. It has been found that a genetically engineered vaccine produced by an *E. coli*, adenovirus, or baculovirus expression system can make up for the deficiency of conventional commercial vaccines, such as attenuation or inactivation, and this has become a trend for vaccine research and development. The baculovirus expression system is a safe and effective vector capable of simultaneously expressing antigens of multiple pathogens; based on this, it is possible to develop a combined genetically engineered vaccine that can prevent MPS and PMWS simultaneously.

In the present study, recombinant baculoviruses rvAc-P97R1P46P42-mCherry-Cap-EGFP and rvAc-P97R1P46P42-Cap were constructed, and the humoral and cellular immune effects induced by rvAc-P97R1P46P42-Cap were evaluated in mice and piglets. mCherry and EGFP tags were used to indirectly observe and accurately detect the expression of these proteins, which could be directly observed in an inverted fluorescence microscope (Figure 3). It was proved that they can be displayed on the plasma membrane of Sf9 cells (Figure 4) and the viral envelope (Figure 5). These results imply that the recombinant baculovirus that displayed mCherry-tagged P97R1P46P42 and EGFP-tagged Cap proteins was properly assembled and is capable of infecting Sf9 cells. Modified baculovirus rvAc-P97R1P46P42-Cap can express P97R1P46P42 and Cap proteins and infect cells successfully (Figure 2).

In mouse immunization, the antibody levels against three Mhp antigens from the rvAc-P97R1P46P42-Cap group were significantly higher (*p* < 0.001) than those from the PBS, rvAc-dual, Mhp CV, and PCV2 CV groups at 35 and 42 DAI (Figure 6A–C). This showed that rvAc-P97R1P46P42-Cap could induce strong humoral immune effects against Mhp antigens in mice. The immune levels of anti-P97R1 and anti-P46 were lower than that of anti-P42, which may be due to the fact that although the flexible joint protein GGSG was added, P97R1 and P46 proteins were not sufficiently exposed during the protein folding, which affected their display on baculovirus, thus affecting the immune effect. While a lower IgG level than the PCV2 CV group against Cap protein (Figure 6D) was also found, this may be because the immunogen of Yuankexin^®^ was a virus-like particle (VLP) type of commercial vaccine developed from *E. coli*. As we know, protein immunogens have a better effect on humoral immunity but a lower effect on cellular immunity than other types of immunogens, which may be because glycosylation and spatial folding of proteins are not good in the prokaryotic expression system. Studies have shown that VLPs have a structure similar to natural viruses but do not contain a viral genome. They are more immunogenic than common proteins, so the antibody level of Yuankexin^®^ induction is higher than that of the live virus immunogen. The results of the lymphocyte proliferation assay indicated that rvAc-P97R1P46P42-Cap could induce strong cellular immune effects against Mhp and PCV2 antigens in mice (Figure 6E). This proved that the cellular immune effects induced by proteins developed by the eukaryotic expression system are better than those from *E. coli*. The results of IgG analysis by indirect ELISA against field strains showed that serum from the rvAc-P97R1P46P42-Cap group could react with native proteins from Mhp and PCV2 (Figure 6F,G). The IgG level of the Mhp CV group was much higher than that from the rvAc-P97R1P46P42-Cap group (Figure 6F), which may be because the expression of antigen proteins in the Mhp 168 strain is similar to that in the strain used in RespiSure^®^ ONE.

In piglet immunization, IgG levels against rP97R1, rP46, rP42, and Cap from the rvAc-P97R1P46P42-Cap group at 28 DAI were significantly higher (*p* < 0.001) than those from the rvAc-dual group (Figure 7A–D), indicating that rvAc-P97R1P46P42-Cap could induce a humoral immune response in piglets. The cytokine IFN-γ produced by CD8+ cells and some CD4+ cells (especially Th1 cells) can enhance the activity of Th1 cells and promote cellular immune response; the Th2 type immune response is mainly manifested by the secretion of cytokine IL-4 and the production of specific antibodies [24,25]. Significantly increased cytokine levels (*p* < 0.001) indicated that rvAc-P97R1P46P42-Cap could induce a Th1/Th2 mixed immune response (Figure 7E,F). Two commercial vaccines showed effective immunogenicity in this experiment, with higher IgG and cytokine levels than the rvAc-P97R1P46P42-Cap group. The effectiveness of the Mhp commercial vaccine was contrary to the results in immune experiments with mice. In the analysis of total IgG levels and stimulation values in mouse immunization, sera and lymphocytes from the Mhp CV group did not react with all the antigens. Similar results also appeared in other studies. Many other researchers found that Mhp commercial vaccines are not effective in mouse immunization experiments [9,10,26,27,28]. However, similar commercial vaccines have been proved to be safe and effective in experimental or field conditions of piglet immunization [7,29,30,31]. The types of vaccines used in these studies are inactivated or attenuated vaccines, which is similar to the inactivated Mhp vaccine used in this study, while the PCV2 vaccine used in this study is a protein-type vaccine, so it works well in both mouse and pig experiments. This may be because the body’s reaction to these inactivated or attenuated vaccines in the corresponding animal was considered in the vaccine composition and production process, and the species difference between animals may affect the responsiveness to the vaccines, so they will not work well for animals of other species. However, researchers have not yet studied whether conventional inactivated and attenuated vaccines have an immune-protective effect in mice. Therefore, evaluating the efficacy and safety of the vaccine must be based on immune experiments with the corresponding animals, although mice are the main experimental animals used at present.

## 4. Materials and Methods

### 4.1. Selection of Coding Sequences and Gene Design and Construction of the Baculovirus Expression Vector

According to previous studies of Mhp and PCV2, coding DNA sequences (CDS) of P97R1 (MHP168_110), P46 (MHP168_522), and P42 (MHP168_069) from Mhp strain 168 (GenBank CP002274.1) were used to construct the chimeric protein antigen; Cap protein (deleting the nuclear localization signal of 41 amino acid residues) from PCV2 isolate strain ShanDong3-2016 (KY656098.1) was used as another antigen (Table 1) [9,15,32]. EGFP and mCherry were used as tag proteins for detection and were cloned from vectors pEGFP-N1 and pmCherry-N1 (Clontech, Otsu, Shiga, Japan), respectively. SPgp64 (GenBank AFO10080.1) and TMDgp64 (GenBank CAA24524.1) were located upstream of the target protein and downstream of the tag protein, respectively.

Flexible linker GGSG (GlyGlySerGly) was inserted between each chimeric subunit protein of P97R1P46P42 to achieve proper folding. The fusion gene fragment SPgp64-P97R1P46P42-mCherry-TMDgp64-6×His was inserted into the baculovirus expression vector pFastBac dual plasmid (Invitrogen, Carlsbad, CA, USA) to generate pFastBac dual-P97R1P46P42-mCherry-Cap-EGFP. SPgp64-Cap-EGFP-TMDgp64-6×His was inserted into pFastBac dual to generate pFastBac dual-P97R1P46P42-Cap.

### 4.2. Culture of Spodoptera Frugiperda Cells (Sf9) and Preparation of Pathogens (PCV2 and Mhp)

The Sf9 cells (laboratory preserved) were cultured at 27 °C in serum-free Sf-900 Ⅱ SFM medium (Gibco, Grand Island, NY, USA), which was used for the proliferation of recombinant baculovirus and gene expression. The porcine kidney cell line (PK-15, provided by Ebvac, Hangzhou, China), free of PCV2 contamination, was maintained in Dulbecco’s Modified Eagle Medium (DMEM, Gibco) supplemented with 10% (*v*/*v*) heat-inactivated fetal bovine serum (FBS; Gibco), 100 U/mL of penicillin, and 100 μg/mL of streptomycin. The ZJ/C strain of PCV2 (provided by Ebvac, Hangzhou, China) was propagated and titrated in PK-15 cells following the standard protocol [33]. The 168 strain of Mhp (purchased from Nanjing Tech-bank Bio-industry Co., Ltd, Nanjing, China) was cultivated in Friis medium using a previously described method [34].

### 4.3. Preparation of Recombinant Bacmid DNA and Proliferation of the Recombinant Baculovirus

Recombinant plasmid pFastBac dual-P97R1P46P42-mCherry-Cap-EGFP and plasmid pFastBac dual (negative control) were transformed into competent DH10 Bac *E. coli* (Invitrogen). Blue/white selection was used to get the positive colonies, and the inserted genes were detected by gene sequencing. Bacmid DNAs of positive and negative colonies were named dual-P97R1P46P42-mCherry-Cap-EGFP and dual, respectively. The recombinant bacmid DNAs of dual-P97R1P46P42-mCherry-Cap-EGFP and dual were transfected into Sf9 cells according to the user manual (Fugene 6, Promega, Madison, WI, USA). Cell fluorescence was observed after incubation for 72 h, 96 h, and 120 h using an inverted fluorescence microscope (Nikon Eclipse TE2000-U, Tokyo, Japan). After 120 h, recombinant viruses were selected and purified by three rounds of plaque isolation and titered by plaque assay. The high virus titer stocks were named rvAc-P97R1P46P42-mCherry-Cap-EGFP and rvAc-dual. Procedures for the preparation of recombinant bacmid DNA and the construction of recombinant baculovirus were performed according to the Bac-to-Bac^®^ Baculovirus Expression System user manual (Invitrogen).

### 4.4. Indirect Immunofluorescence and Immunogold Electron Microscopy Assays

To visualize the location of chimeric P97R1P46P42 and Cap proteins in infected Sf9 cells, recombinant baculovirus rvAc-P97R1P46P42-mCherry-Cap-EGFP and rvAc-dual (multiplicity of infection (MOI): 10) were transduced to Sf9 cells. Experimental procedures were conducted according to a previous study [15]. The antibodies used in indirect immunofluorescence assay were as follows: equal-volume mixed anti-EGFP monoclonal antibody (1:1000 dilution; Beyotime Biotechnology, Shanghai, China), mouse anti-mCherry monoclonal antibody (1:1000 dilution; Bioss, Beijing, China), and Alexa Fluor 555-conjugated donkey anti-mouse IgG (secondary antibody, 1:500 dilution; Beyotime Biotechnology). Fluorescent images were photographed with a confocal microscope (Leica TCS SPE, Wetzlar, Germany).

To visualize the location of chimeric P97R1P46P42 and Cap proteins in the baculovirus envelope, rvAc-P97R1P46P42-mCherry-Cap-EGFP and rvAc-dual were reproduced as mentioned above. The virus supernatant was purified by sucrose gradient ultracentrifugation following standard protocol [35]. For immunogold electron microscopy assay, antibodies used were as follows: mouse anti-EGFP monoclonal antibody (1:100 dilution; Beyotime Biotechnology), mouse anti-mCherry monoclonal antibody (1:100 dilution; Bioss), and Protein A-Gold (secondary antibody, 1:100 dilution; Bioss). The grids were stained with 3% phosphotungstic acid and photographed with a transmission electron microscope at 80 kv (Hitachi H-7650, Tokyo, Japan).

### 4.5. Transformation of the Vector and Baculovirus for Animal Immune Experiments

In order to get the protein without fluorescent tags, another vector, pFastBac dual-P97R1P46P42-Cap, was generated which removed tag proteins EGFP and mCherry. Recombinant baculovirus rvAc-P97R1P46P42-Cap was obtained by the same method mentioned above, and rvAc-dual was used as a negative control. The infected Sf9 cell lysate and purified baculovirus rvAc-P97R1P46P42-Cap were identified by Western blot according to a previous study [36]. Antibodies used were as follows: rabbit anti-rP97R1, rabbit anti-rP46, and rabbit anti-rP42 polyclonal antibodies (1:1000 dilution, laboratory preserved), rabbit anti-Cap polyclonal antibodies (1:5000 dilution; Bioss), and monoclonal mouse anti-6 × His (Beyotime Biotechnology).

### 4.6. Mouse Vaccination

Thirty healthy female BALB/c mice (6–8 weeks old, purchased from Laboratory Animal Research Center of Zhejiang Chinese Medical University, Hangzhou, China) were randomly divided into five groups with six mice per group (Table 2). Subcutaneous injection with vaccine components was carried out at 14-day intervals. The dose of commercial vaccines was based on related research [9,37]. Serum samples were collected from the retro-orbital sinus at 0, 14, 28, 35, and 42 days after immunization (DAI). Three mice from each group were sacrificed by cardiac puncture at 35 and 42 DAI and the spleen was separated for lymphocyte proliferation assay. Mouse experiments were conducted in accordance with the recommendations of the Ethics Committee for Animal Experimentation of the College of Life Sciences of Zhejiang Sci-Tech University (Code: 20180401, Date: 2, April, 2018).

### 4.7. IgG Antibody Analysis

The antibody levels were determined using indirect enzyme-linked immunosorbent assay (ELISA) according to a previous study [36]. In detail, microtiter plates were coated with subunit protein antigens (rP97R1, rP46, rP42, and Cap at 1 µg/mL, laboratory preserved, and stored at −80 °C), followed by incubation with diluted serum samples (diluted 1:100) from 0, 14, 21, 35, and 42 DAI, then horseradish peroxidase (HRP)-conjugated goat anti-mouse IgG was added (diluted 1:1000; Beyotime). All ELISA reactions were assessed in triplicate, and absorbance was determined at an optical density (OD) of 450 nm using an enzyme-labeled meter (Bio-Tek, Winooski, VT, USA).

### 4.8. Lymphocyte Proliferation Assay

Experimental steps were as previously described [15], and the lymphocyte proliferation assay was performed in triplicate. In detail, 100 μL RPMI-1640 (Gibco) containing 2 μg rP97R1, 2 μg rP46, 2 μg rP42, and 2 μg Cap was used as a stimulator, and RPMI-1640 and concanavalin A (5 μg/mL; Sigma, St. Louis, MO, USA) were added as negative and positive controls, respectively. Cell proliferation of splenic lymphocytes was detected by standard MTT assay (5 mg/mL; Sigma). The OD value was determined at 490 nm and stimulation was calculated as follows: stimulation (%) = mean OD of antigen-stimulated cells (rP97R1, rP46, rP42, and Cap added)/mean OD of unstimulated cells (RPMI-1640 added)/mean OD of concanavalin-A-stimulated cells.

### 4.9. Reactivity with Native Proteins

To verify whether the antibodies induced by mouse immunization were able to recognize native proteins from Mhp and PCV2, pooled serum samples from 42 DAI (diluted 1:100) were evaluated by indirect ELISA. Each well of the microtiter plates was coated with 100 ng crude extract of the Mhp 168 strain and PCV2 ZJ/C strain. The plates were incubated at 4 °C overnight, then incubated at –80 °C for 2 h and thawed at room temperature for 30 min; the detection process is described in detail in [38].

### 4.10. Piglet Vaccination

To determine the immunogenicity of the recombinant baculovirus in piglets, nine 4-week-old male domestic piglets (7–8 kg, purchased from Laboratory Animal Research Center of Zhejiang Chinese Medical University, Hangzhou, China) from an Mhp- and PCV2-free herd were randomly divided into three groups with three animals per group; detailed information can be found in Table 3. The dose of baculovirus was based on related studies [15], and the dose of commercial vaccines was according to the products’ user manuals. Pig experiments were conducted in accordance with the recommendations of the Ethics Committee for Animal Experimentation of the College of Life Sciences of Zhejiang Sci-Tech University (Code: 20190102, Date: 10^th^, Jan, 2019). Serum samples were collected at 0 and 28 DAI. Systemic IgG was measured by indirect ELISA using the same procedure described above, and HRP-conjugated rabbit anti-piglet IgG (1:5000; Bioss) was used as the secondary antibody. The cellular response was analyzed by detection of IL-4 and IFN-γ cytokines in serum according to other related research [39,40,41]; in this case, commercial pig cytokine double antibody sandwich ELISA kits (Feiya Biological Technology, Yancheng, China) were used. 

### 4.11. Statistical Analysis

Statistical analysis was conducted using GraphPad Prism^®^ Version 6 for Windows (GraphPad Software, San Diego, CA, USA). Data were expressed as mean ± SD. Bonferroni post-testing was used to compare the immune responses between groups. Data were considered significantly different at *p* < 0.05.

## 5. Conclusions

In summary, this report described the production, identification, and immunogenicity of a recombinant baculovirus that displays chimeric antigen proteins of Mhp and PCV2. The results indicate that this recombinant baculovirus was able to induce humoral and cellular effects in mice and piglets, and moreover, it represents a potential combined vaccine candidate against both MPS and PMWS. Further investigation will focus on a co-infection challenge with these two diseases in piglets to determine clinical protection under experimental and field conditions.

## Figures and Tables

**Figure 1 ijms-20-04425-f001:**
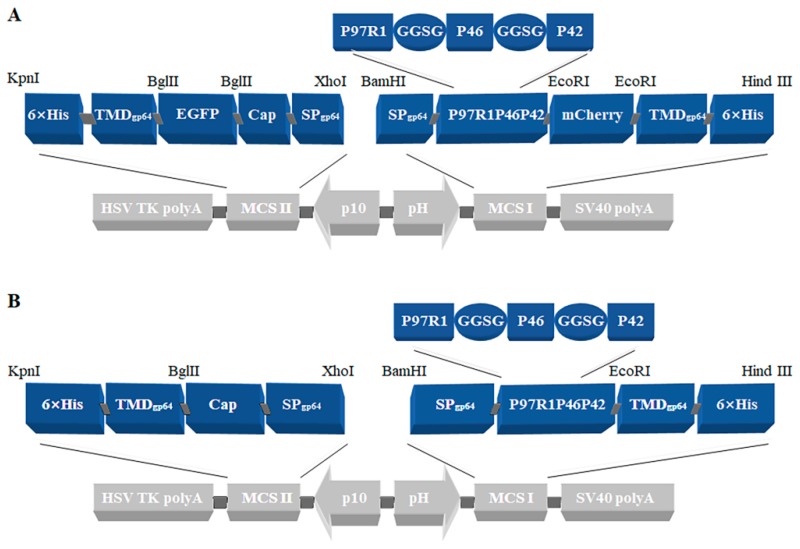
Map of recombinant vectors: (**A**) pFastBac dual-P97R1P46P42-mCherry-Cap-EGFP; (**B**) pFastBac dual-P97R1P46P42-Cap.

**Figure 2 ijms-20-04425-f002:**
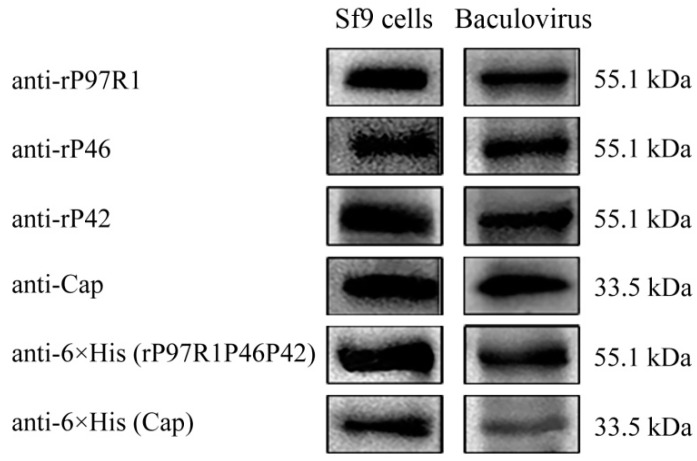
Detection of P97R1P46P42 and Cap expression in rvAc-P97R1P46P42-Cap-infected Sf9 cells and concentrated baculovirus rvAc-P97R1P46P42-Cap.

**Figure 3 ijms-20-04425-f003:**
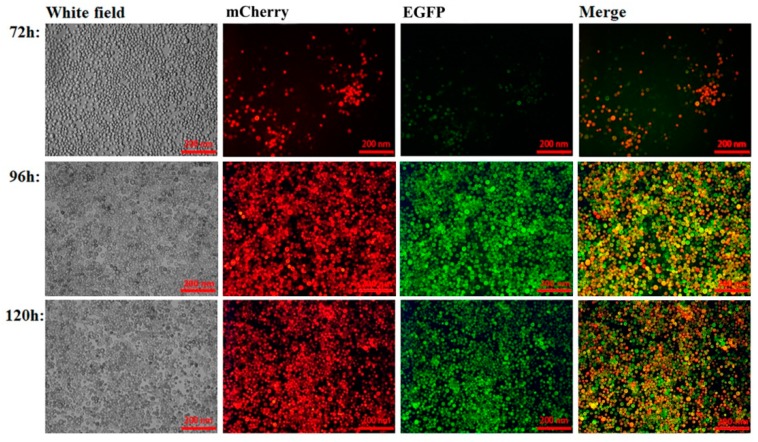
Visualization using mCherry and EGFP fluorescent proteins of rvAc-P97R1P46P42-mCherry-Cap-EGFP-infected Sf9 cells; both of the antigens can be produced by the recombinant baculovirus.

**Figure 4 ijms-20-04425-f004:**
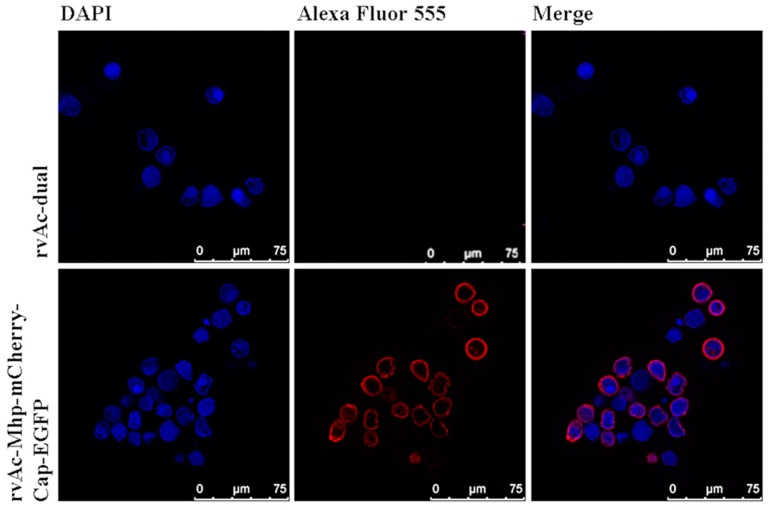
Confocal microscopy analysis showed the P97R1P46P42 and Cap proteins anchored on the plasma membrane of infected Sf9 cells.

**Figure 5 ijms-20-04425-f005:**
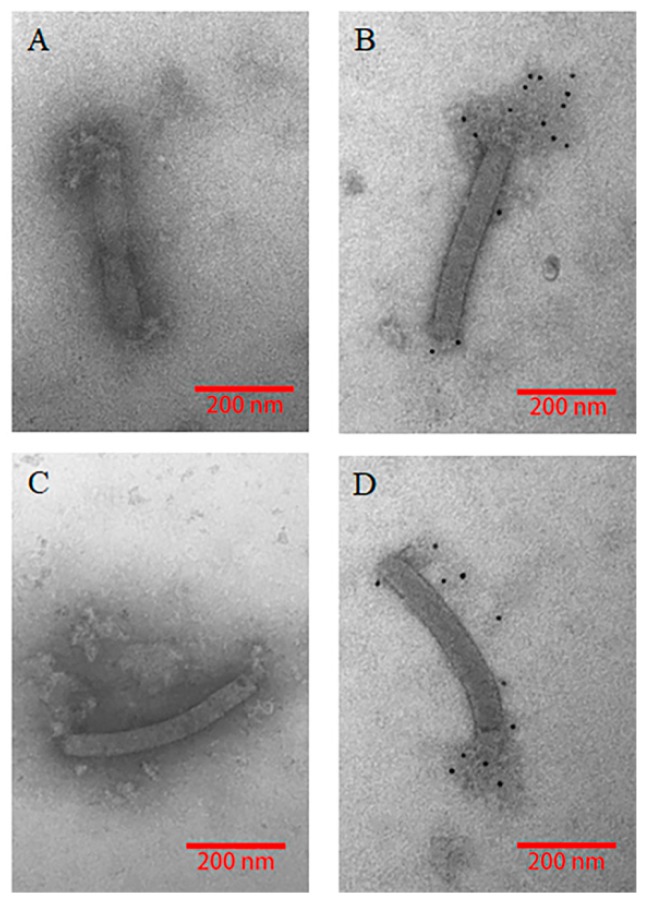
Immunogold electron micrographs of purified baculoviruses rvAc-P97R1P46P42-mCherry-Cap-EGFP and rvAc-dual. (**A**) rvAc-dual incubated with mCherry monoclonal antibody; (**B**) rvAc-P97R1P46P42-mChery-Cap-EGFP incubated with mCherry monoclonal antibody; (**C**) rvAc-dual incubated with EGFP monoclonal antibody; (**D**) rvAc-P97R1P46P42-mChery-Cap-EGFP incubated with EGFP monoclonal antibody. On the surface of recombinant baculovirus incubated with mCherry and EGFP monoclonal antibodies, about 10 nm diameter colloidal gold particles were clustered in the envelope region, and no colloidal gold particles were found on the wild-type baculovirus surface.

**Figure 6 ijms-20-04425-f006:**
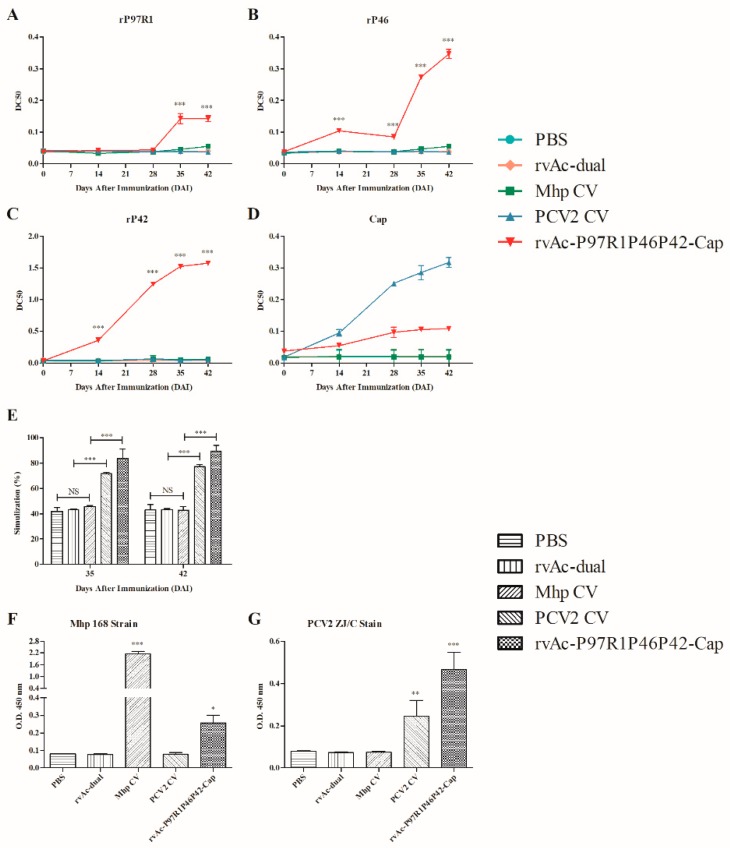
Analysis of mouse immunization by indirect ELISA and lymphocyte proliferative experiments. Analysis of the immunoglobulin G (IgG) response induced by mouse immunization determined by indirect ELISA with four recombinant proteins (**A**–**D**). (**A**) Total IgG level against rP97R1; (**B**) Total IgG level against rP46; (**C**) Total IgG level against rP42; (**D**) Total IgG level against Cap. The *Y* axis represents the mean EC50 (concentration for 50% of maximal effect) of serum samples collected at 0, 14, 28, 35, and 42 days after immunization (DAI) in each group. *** *p* < 0.001, significantly different from the PBS, rvAc-dual, Mhp CV, and PCV2 CV groups (Bonferroni test). Lymphocyte proliferative experiment results (**E**). The *Y* axis represents the stimulation of splenic lymphocyte samples collected at 35 and 42 DAI. NS: not significantly different; *** *p* < 0.001, significantly different (Bonferroni test). Analysis of the IgG response induced by mouse immunization determined by indirect ELISA against Mhp 168 and PCV2 ZJ/C strains (**F**,**G**). (**F**) Total IgG level against Mhp 168 strain; (**G**) Total IgG level against PCV2 ZJ/C strain. The *Y* axis represents the mean OD 450 of serum samples collected at 42 DAI in each group. * *p* < 0.05, ** *p* < 0.01, and *** *p* < 0.001, significantly different from the PBS and rvAc-dual groups (Bonferroni test). All analyses were performed in triplicate, and the error bars demonstrate standard deviations (SD).

**Figure 7 ijms-20-04425-f007:**
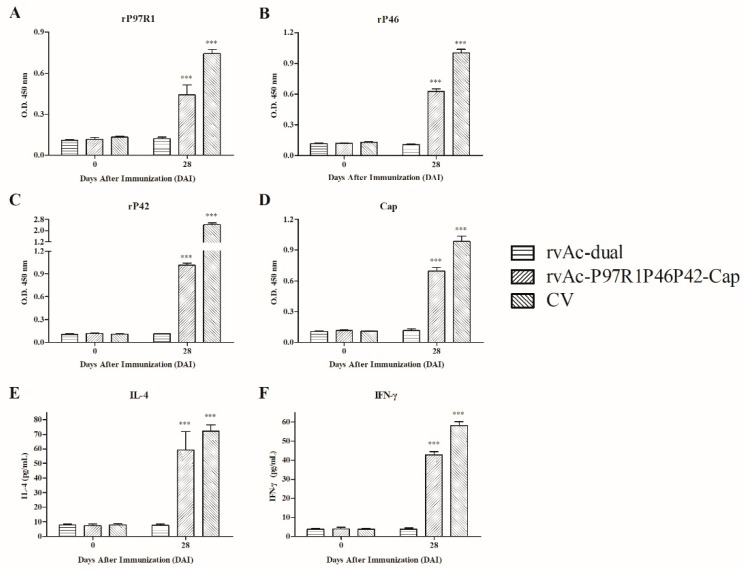
Analysis of immunogenicity in piglets. IgG response in piglet immunization against four recombinant proteins (**A**–**D**). (**A**) Total IgG level against rP97R1; (**B**) Total IgG level against rP46; (**C**) Total IgG level against rP42; (**D**) Total IgG level against Cap. The *Y* axis represents the mean OD 450 of serum samples collected at 0 and 28 DAI. Quantitative analysis of interleukin 4 (IL-4) and interferon gamma (IFN-γ) levels in sera from immunized piglets (**E**,**F**). (**E**) Content of cytokine IL-4; (**F**) Content of cytokine IFN-γ. The *Y* axis represents the mean content of cytokine in serum samples collected at 0 and 28 DAI. All analyses were performed in triplicate, and the error bars demonstrate standard deviations (SD). *** *p* < 0.001, significantly different from the OD value of the rvAc-dual group (Bonferroni test).

**Table 1 ijms-20-04425-t001:** Antigens selected in this experiment.

Protein	NCBI Accession Number	Features/Function	Original Amino Acid Length (aa)	Selected Fragment (aa)
Mhp P97R1	ADQ90328.1	Cilium adhesin	1082	788–915
Mhp P46	ADQ90718.1	46 kDa surface antigen	120	323–419
Mhp P42	ADQ90292.1	Chaperone protein DnaK	622	434–600
PCV2 Cap	ARW74078.1	Capsid protein	234	42–234

**Table 2 ijms-20-04425-t002:** Mouse vaccination strategies.

Group	Formulation	Immunization Time Points (DAI)	Dose
1, PBS	PBS	0, 14, 28	100 μL
2, rvAc-dual	rvAc-dual	0, 14, 28	10^8^ PFU
3, Mhp CV	RespiSure^®^ ONE commercial vaccine	0, 14, 28	100 μL
4, PCV2 CV	Yuankexin^®^ commercial vaccine	0, 14, 28	100 μL
5, rvAc-P97R1P46P42-Cap	rvAc-P97R1P46P42-Cap	0, 14, 28	10^8^ PFU

**Table 3 ijms-20-04425-t003:** Piglet vaccination strategies.

Group	Formulation	Immunization Time Points (DAI)	Dose
1, rvAc-dual	rvAc-dual	0, 14	10^9^ PFU
2, rvAc-P97R1P46P42-Cap	rvAc-P97R1P46P42-Cap	0, 14	10^9^ PFU
3, CV	RespiSure^®^ ONE and Yuankexin^®^	0, 14	Take 1 mL of each and mix

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
