# Peer review of "Development of a Combined Genetic Engineering Vaccine for Porcine Circovirus Type 2 and Mycoplasma Hyopneumoniae by a Baculovirus Expression System"

_ijms, 2019, doi:10.3390/ijms20184425_

Round 1
Reviewer 1 Report
This current revised MS addressed most of my comments except that
1) Fig 2 is quite suspicious in its current format. Some improvement could be done in the next round of revision. Protein samples from cells and viruses against the same Ab could be tested together in one membrane. Importantly, molecular weight (kDa) should have been marked in the figure.
2) Why did the authors provide images in different magnification rate? 96 hpi--> 400 nm. Please replace this.
3) Some raw data for Fig. 5 is missing. Please provide all the data set. The low number of BV presented could be improved by the ultra-concentration method. It is fine to keep the data as it is. However, some discussion should be included because preparing a high titer BV is always demanded before immunization in any animal models and for long-term storage.
4) The resolution for Figs 6&7 is bad. Please improve also the data presentation. Since all the sub-figures use the same labels/symbols, it is ok to make the figure more compact.
Author Response
1) Fig 2 is quite suspicious in its current format. Some improvement could be done in the next round of revision. Protein samples from cells and viruses against the same Ab could be tested together in one membrane. Importantly, molecular weight (kDa) should have been marked in the figure.
We have added the molecular weight information in the Figure 2. Protein samples from cells and viruses could be detected by the same antibody, but due to the protein content in cells is much higher than that in the viruses, which cause the different exposure time in WB experiment. Under the condition of maintaining the same sample ratio, it is difficult to show two bands with better effect on the same membrane. If it is necessary to test proteins from cells and viruses in one membrane, we are willing to run a WB again. However, the production of the virus and cell samples need some time, and this revision is only given for a period of one day, so we are very sorry to cannot provide the results immediately.
2) Why did the authors provide images in different magnification rate? 96 hpi--> 400 nm. Please replace this.
The magnification rate of figures at 96 h was different from 72 h and 120 h. As the suggestion, we have replaced these figures (Figure 3), raw data of fluorescent pictures for this revision were also provided (Supplementary Files).
3) Some raw data for Fig. 5 is missing. Please provide all the data set. The low number of BV presented could be improved by the ultra-concentration method. It is fine to keep the data as it is. However, some discussion should be included because preparing a high titer BV is always demanded before immunization in any animal models and for long-term storage.
We have provided original data of Fig 5 (Supplementary Files). As you said, ultra-concentration can indeed improve the titer of the virus. However, while the virus is concentrated, the salt crystal and some other impurities in the buffer where the viruses located will also be concentrated, resulting in poor spread of phosphotungstic acid, and causing a deep background of the figure. Therefore, we used a lower concentration of the virus solution when making the immune electron microscope, and the purpose is to show the display of the target protein more clearly. For long-term storage and immunization, we have controlled the titer of the virus in 109 PFU/mL by purification and concentration. Relevant description has been added to the manuscript (Result section, line 80-81, page 2).
4) The resolution for Figs 6&7 is bad. Please improve also the data presentation. Since all the sub-figures use the same labels/symbols, it is ok to make the figure more compact.
Thank you for your suggestion. We have combined all same labels/symbols, and made the figure more compact (Fig 6 and Fig 7).
Reviewer 2 Report
Authors have made extensive revisions of their manuscript as required by reviewers. The manuscript can be published in its present form.
Author Response
Thanks very much and we are feel so warm for your suggestions.